# PRINCIPLED WEIGHT INITIALIZATION FOR HYPERNETWORKS

**Oscar Chang, Lampros Flokas, Hod Lipson**
Columbia University
New York, NY 10027
{oscar.chang, lf2540, hod.lipson}@columbia.edu

## ABSTRACT

Hypernetworks are meta neural networks that generate weights for a main neural network in an end-to-end differentiable manner. Despite extensive applications ranging from multi-task learning to Bayesian deep learning, the problem of optimizing hypernetworks has not been studied to date. We observe that classical weight initialization methods like Glorot & Bengio (2010) and He et al. (2015), when applied directly on a hypernet, fail to produce weights for the mainnet in the correct scale. We develop principled techniques for weight initialization in hypernets, and show that they lead to more stable mainnet weights, lower training loss, and faster convergence.

## 1 INTRODUCTION

Meta-learning describes a broad family of techniques in machine learning that deals with the problem of learning to learn. An emerging branch of meta-learning involves the use of *hypernetworks*, which are meta neural networks that generate the weights of a main neural network to solve a given task in an end-to-end differentiable manner. Hypernetworks were originally introduced by Ha et al. (2016) as a way to induce weight-sharing and achieve model compression by training the same meta network to learn the weights belonging to different layers in the main network. Since then, hypernetworks have found numerous applications including but not limited to: weight pruning (Liu et al., 2019), neural architecture search (Brock et al., 2017; Zhang et al., 2018), Bayesian neural networks (Krueger et al., 2017; Ukai et al., 2018; Pawlowski et al., 2017; Henning et al., 2018; Deutsch et al., 2019), multi-task learning (Pan et al., 2018; Shen et al., 2017; Klocek et al., 2019; Serrà et al., 2019; Meyerson & Miikkulainen, 2019), continual learning (von Oswald et al., 2019), generative models (Suarez, 2017; Ratzlaff & Fuxin, 2019), ensemble learning (Kristiadi & Fischer, 2019), hyperparameter optimization (Lorraine & Duvenaud, 2018), and adversarial defense (Sun et al., 2017).

Despite the intensified study of applications of hypernetworks, the problem of optimizing them to this day remains significantly understudied. In fact, even the problem of initializing hypernetworks has not been studied. Given the lack of principled approaches, prior work in the area is mostly limited to ad-hoc approaches based on trial and error (c.f. Section 3). For example, it is common to initialize the weights of a hypernetwork by sampling a "small" random number. Nonetheless, these ad-hoc methods do lead to successful hypernetwork training primarily due to the use of the Adam optimizer (Kingma & Ba, 2014), which has the desirable property of being invariant to the scale of the gradients. However, even Adam will not work if the loss diverges (i.e. overflow) at initialization, which will happen in sufficiently big models. The normalization of badly scaled gradients also results in noisy training dynamics where the loss function suffers from bigger fluctuations during training compared to vanilla stochastic gradient descent (SGD). Wilson et al. (2017); Reddi et al. (2018) showed that while adaptive optimizers like Adam may exhibit lower training error, they fail to generalize as well to the test set as non-adaptive gradient methods. Moreover, Adam incurs a computational overhead and requires 3X the amount of memory for the gradients compared to vanilla SGD.

Small random number sampling is reminiscent of early neural network research (Rumelhart et al., 1986) before the advent of classical weight initialization methods like Xavier init (Glorot & Bengio, 2010) and Kaiming init (He et al., 2015). Since then, a big lesson learned by the neural network

optimization community is that architecture specific initialization schemes are important to the robust training of deep networks, as shown recently in the case of residual networks (Zhang et al., 2019). In fact, weight initialization for hypernetworks was recognized as an outstanding open problem by prior work (Deutsch et al., 2019) that had questioned the suitability of classical initialization methods for hypernetworks.

**Our results** We show that when classical methods are used to initialize the weights of hypernetworks, they fail to produce mainnet weights in the correct scale, leading to exploding activations and losses. This is because classical network weights transform one layer's activations into another, while hypernet weights have the added function of transforming the hypernet's activations into the mainnet's weights. Our solution is to develop principled techniques for weight initialization in hypernetworks based on variance analysis. The hypernet case poses unique challenges. For example, in contrast to variance analysis for classical networks, the case for hypernetworks can be asymmetrical between the forward and backward pass. The asymmetry arises when the gradient flow from the mainnet into the hypernet is affected by the biases, whereas in general, this does not occur for gradient flow in the mainnet. This underscores again why architecture specific initialization schemes are essential. We show both theoretically and experimentally that our methods produce hypernet weights in the correct scale. Proper initialization mitigates exploding activations and gradients or the need to depend on Adam. Our experiments reveal that it leads to more stable mainnet weights, lower training loss, and faster convergence.

Section 2 briefly covers the relevant technical preliminaries, and Section 3 reviews problems with the ad-hoc methods currently deployed by hypernetwork practitioners. We derive novel weight initialization formulae for hypernetworks in Section 4, empirically evaluate our proposed methods in Section 5, and finally conclude in Section 6.

## 2 PRELIMINARIES

**Definition.** *A **hypernetwork** is a meta neural network $H$ with its own parameters $\phi$ that generates the weights of a main network $\theta$ from some embedding $e$ in a differentiable manner: $\theta = H_\phi(e)$. Unlike a classical network, in a hypernetwork, the weights of the main network are not model parameters. Thus the gradients $\Delta\theta$ have to be further backpropagated to the weights of the hypernetwork $\Delta\phi$, which is then trained via gradient descent $\phi_{t+1} = \phi_t - \lambda\Delta\phi_t$.*

This fundamental difference suggests that conventional knowledge about neural networks may not apply directly to hypernetworks and novel ways of thinking about weight initialization, optimization dynamics and architecture design for hypernetworks are sorely needed.

### 2.1 RICCI CALCULUS

We propose the use of Ricci calculus, as opposed to the more commonly used matrix calculus, as a suitable mathematical language for thinking about hypernetworks. Ricci calculus is useful because it allows us to reason about the derivatives of higher-order tensors with notational ease. For readers not familiar with the index-based notation of Ricci calculus, please refer to Laue et al. (2018) for a good introduction to the topic written from a machine learning perspective.

For a general nth-order tensor $T^{i_1,\cdots,i_k,\cdots,i_n}$, we use $\mathrm{d}_{i_k}$ to refer to the dimension of the index set that $i_k$ is drawn from. We include explicit summations where the relevant expressions might be ambiguous, and use Einstein summation convention otherwise. We use square brackets to denote different layers for added clarity, so for example $W[t]$ denotes the $t$-th weight layer.

### 2.2 XAVIER INITIALIZATION

Glorot & Bengio (2010) derived weight initialization formulae for a feedforward neural network by conducting a variance analysis over activations and gradients. For a linear layer $y^i = W^i_j x^j + b^i$, suppose we make the following **Xavier Assumptions** at initialization: (1) The $W^i_j$, $x^j$, and $b^i$ are all independent of each other. (2) $\forall i, j : \mathbb{E}[W^i_j] = 0$. (3) $\forall j : \mathbb{E}[x^j] = 0$. (4) $\forall i : b^i = 0$.

Then, $\mathbb{E}[y^i] = 0$ and $\text{Var}(y^i) = \text{d}_j \text{Var}(W^i_j)\text{Var}(x^j)$. To keep the variance of the output and input activations the same, i.e. $\text{Var}(y^i) = \text{Var}(x^j)$, we have to sample $W^i_j$ from a distribution whose variance is equal to the reciprocal of the fan-in: $\text{Var}(W^i_j) = \frac{1}{\text{d}_j}$.

If analogous assumptions hold for the backward pass, then to keep the variance of the output and input gradients the same, we have to sample $W^i_j$ from a distribution whose variance is equal to the reciprocal of the fan-out: $\text{Var}(W^i_j) = \frac{1}{\text{d}_i}$.

Thus, the forward pass and backward pass result in symmetrical formulae. Glorot & Bengio (2010) proposed an initialization based on their harmonic mean: $\text{Var}(W^i_j) = \frac{2}{\text{d}_j + \text{d}_i}$.

In general, a feedforward network is non-linear, so these assumptions are strictly invalid. But odd activation functions with unit derivative at $0$ results in a roughly linear regime at initialization.

## 2.3 KAIMING INITIALIZATION

He et al. (2015) extended Glorot & Bengio (2010)'s analysis by looking at the case of ReLU activation functions, i.e. $y^i = W^i_j \text{ReLU}(x^j) + b^i$. We can write $z^j = \text{ReLU}(x^j)$ to get

$$\text{Var}(y^i) = \sum_j \mathbb{E}[(z^j)^2]\text{Var}(W^i_j) = \sum_j \frac{1}{2}\,\mathbb{E}[(x^j)^2]\text{Var}(W^i_j) = \frac{1}{2}\,\text{d}_j\text{Var}(W^i_j)\text{Var}(x^j).$$

This results in an extra factor of $2$ in the variance formula. $W^i_j$ have to be symmetric around $0$ to enforce Xavier Assumption 3 as the activations and gradients propagate through the layers. He et al. (2015) argued that both the forward or backward version of the formula can be adopted, since the activations or gradients will only be scaled by a depth-independent factor. For convolutional layers, we have to further divide the variance by the size of the receptive field.

'Xavier init' and 'Kaiming init' are terms that are sometimes used interchangeably. Where there might be confusion, we will refer to the forward version as fan-in init, the backward version as fan-out init, and the harmonic mean version as harmonic init.

## 3 REVIEW OF CURRENT METHODS

In the seminal Ha et al. (2016) paper, the authors identified two distinct classes of hypernetworks: dynamic (for recurrent networks) and static (for convolutional networks). They proposed Orthogonal init (Saxe et al., 2013) for the dynamic class, but omitted discussion of initialization for the static class. The static class has since proven to be the dominant variant, covering all kinds of non-recurrent networks (not just convolutional), and thus will be the central object of our investigation.

Through an extensive literature and code review, we found that hypernet practitioners mostly depend on the Adam optimizer, which is invariant to and normalizes the scale of gradients, for training and resort to one of four weight initialization methods:

M1 Xavier or Kaiming init (as found in Pawlowski et al. (2017); Balazevic et al. (2018); Serrà et al. (2019); von Oswald et al. (2019)).

M2 Small random values (as found in Krueger et al. (2017); Lorraine & Duvenaud (2018)).

M3 Kaiming init, but with the output layer scaled by $\frac{1}{10}$ (as found in Ukai et al. (2018)).

M4 Kaiming init, but with the hypernet embedding set to be a suitably scaled constant (as found in Meyerson & Miikkulainen (2019)).

M1 uses classical neural network initialization methods to initialize hypernetworks. This fails to produce weights for the main network in the correct scale. Consider the following illustrative example of a one-layer linear hypernet generating a linear mainnet with $T + 1$ layers, given embeddings sampled from a standard normal distribution and weights sampled entry-wise from a zero-mean

distribution. We leave the biases out for now, and assume the input data $x[1]$ is standardized.

$$x[t+1]^{i_{t+1}} = W[t]^{i_{t+1}}_{i_t} x[t]^{i_t}, \qquad W[t]^{i_{t+1}}_{i_t} = H[t]^{i_{t+1}}_{i_t k_t} e[t]^{k_t}, \qquad 1 \le t \le T.$$

$$\mathrm{Var}(x[T+1]^{i_{t+1}}) = \mathrm{Var}(x[1]^{i_1}) \prod_{t=1}^{T} \mathrm{d}_{i_t} \mathrm{Var}(W[t]^{i_{t+1}}_{i_t}) = \mathrm{Var}(x[1]^{i_1}) \prod_{t=1}^{T} \mathrm{d}_{i_t} \mathrm{d}_{k_t} \mathrm{Var}(H[t]^{i_{t+1}}_{i_t k_t}). \tag{1}$$

In this case, if the variance of the weights in the hypernet $\mathrm{Var}(H[t]^{i_{t+1}}_{i_t k_t})$ is equal to the reciprocal of the fan-in $\mathrm{d}_{k_t}$, then the variance of the activations $\mathrm{Var}(x[T+1]^{i_{t+1}}) = \prod_{t=1}^{T} \mathrm{d}_{i_t}$ explodes. If it is equal to the reciprocal of the fan-out $\mathrm{d}_{i_t} \mathrm{d}_{i_{t+1}}$, then the activation variance $\mathrm{Var}(x[T+1]^{i_{t+1}}) = \prod_{t=1}^{T} \frac{\mathrm{d}_{k_t}}{\mathrm{d}_{i_{t+1}}}$ is likely to vanish, since the size of the embedding vector is typically small relatively to the width of the mainnet weight layer being generated.

Where the fan-in is of a different scale than the fan-out, the harmonic mean has a scale close to that of the smaller number. Therefore, the fan-in, fan-out, and harmonic variants of Xavier and Kaiming init will all result in activations and gradients that scale exponentially with the depth of the mainnet.

M2 and M3 introduce additional hyperparameters into the model, and the ad-hoc manner in which they work is reminiscent of pre deep learning neural network research, before the introduction of classical initialization methods like Xavier and Kaiming init. This ad-hoc manner is not only inelegant and consumes more compute, but will likely fail for deeper and more complex hypernetworks.

M4 proposes to set the embeddings $e[t]^{k_t}$ to a suitable constant ($\mathrm{d}_{i_t}^{-1/2}$ in this case), such that both $W[t]^{i_{t+1}}_{i_t}$ and $H[t]^{i_{t+1}}_{i_t k_t}$ can seem to be initialized with the same variance as Kaiming init. This ensures that the variance of the activations in the mainnet are preserved through the layers, but the restrictions on the embeddings might not be desirable in many applications.

Luckily, the fix appears simple — set $\mathrm{Var}(H[t]^{i_{t+1}}_{i_t k_t}) = \frac{1}{\mathrm{d}_{i_t} \mathrm{d}_{k_t}}$. This results in the variance of the generated weights in the mainnet $\mathrm{Var}(W[t]^{i_{t+1}}_{i_t}) = \frac{1}{\mathrm{d}_{i_t}}$ resembling conventional neural networks initialized with fan-in init. This suggests a general hypernet weight initialization strategy: initialize the weights of the hypernet such that the mainnet weights approximate classical neural network initialization. We elaborate on and generalize this intuition in Section 4.

## 4   HYPERFAN INITIALIZATION

Most hypernetwork architectures use a linear output layer so that gradients can pass from the mainnet into the hypernet directly without any non-linearities. We make use of this fact in developing methods called *hyperfan-in init* and *hyperfan-out init* for hypernetwork weight initialization based on the principle of variance analysis.

### 4.1   HYPERFAN-IN

**Proposition.** *Suppose a hypernetwork comprises a linear output layer. Then, the variance between the input and output activations of a linear layer in the mainnet $y^i = W^i_j x^j + b^i$ can be preserved using fan-in init in the hypernetwork with appropriately scaled output layers.*

**Case 1. The hypernet generates the weights but not the biases of the mainnet.** The bias in the mainnet is initialized to zero. We can write the weight generation in the form $W^i_j = H^i_{jk} h(e)^k + \beta^i_j$ where $h$ computes all but the last layer of the hypernet and $(H, \beta)$ form the output layer. We make the following **Hyperfan Assumptions** at initialization: **(1)** Xavier assumptions hold for all the layers in the hypernet. **(2)** The $H^i_{jk}$, $h(e)^k$, $\beta^i_j$, $x^j$, and $b^i$ are all independent of each other. **(3)** $\forall i, j, k : \mathbb{E}[H^i_{jk}] = 0$. **(4)** $\mathbb{E}[x^j] = 0$. **(5)** $\forall i : b^i = 0$.

Use fan-in init to initialize the weights for $h$. Then, $\mathrm{Var}(h(e)^k) = \mathrm{Var}(e^l)$. If we initialize $H$ with the formula $\mathrm{Var}(H^i_{jk}) = \frac{1}{\mathrm{d}_j \mathrm{d}_k \mathrm{Var}(e^l)}$ and $\beta$ with zeros, we arrive at $\mathrm{Var}(W^i_j) = \frac{1}{\mathrm{d}_j}$, which is the formula for fan-in init in the mainnet. The Hyperfan assumptions imply the Xavier assumptions

hold in the mainnet, thus preserving the input and output activations.

$$\text{Var}(y^i) = \sum_j \text{Var}(W_j^i)\text{Var}(x^j) = \sum_j \sum_k \text{Var}(H_{jk}^i)\text{Var}(h(e)^k)\text{Var}(x^j)$$

$$= \sum_j \sum_k \frac{1}{\text{d}_j \text{d}_k \text{Var}(e^l)} \text{Var}(e^l)\text{Var}(x^j) = \text{Var}(x^j). \tag{2}$$

**Case 2. The hypernet generates both the weights and biases of the mainnet.** We can write the weight and bias generation in the form $W_j^i = H_{jk}^i h(e[1])^k + \beta_j^i$ and $b^i = G_l^i g(e[2])^l + \gamma^i$ respectively, where $h$ and $g$ compute all but the last layer of the hypernet, and $(H, \beta)$ and $(G, \gamma)$ form the output layers. We modify Hyperfan Assumption 2 so it includes $G_l^i$, $g(e[2])^l$, and $\gamma^i$, and further assume $\text{Var}(x^j) = 1$, which holds at initialization with the common practice of data standardization.

Use fan-in init to initialize the weights for $h$ and $g$. Then, $\text{Var}(h(e[1])^k) = \text{Var}(e[1]^m)$ and $\text{Var}(g(e[2])^l) = \text{Var}(e[2]^n)$. If we initialize $H$ with the formula $\text{Var}(H_{jk}^i) = \frac{1}{2\text{d}_j \text{d}_k \text{Var}(e[1]^m)}$, $G$ with the formula $\text{Var}(G_l^i) = \frac{1}{2\text{d}_l \text{Var}(e[2]^n)}$, and $\beta, \gamma$ with zeros, then the input and output activations in the mainnet can be preserved.

$$\text{Var}(y^i) = \sum_j \left[\text{Var}(W_j^i)\text{Var}(x^j)\right] + \text{Var}(b^i)$$

$$= \sum_j \left[\sum_k \text{Var}(H_{jk}^i)\text{Var}(h(e[1])^k)\text{Var}(x^j)\right] + \sum_l \text{Var}(G_l^i)\text{Var}(g(e[2])^l)$$

$$= \sum_j \left[\sum_k \frac{1}{2\text{d}_j \text{d}_k \text{Var}(e[1]^m)}\text{Var}(e[1]^m)\text{Var}(x^j)\right] + \sum_l \frac{1}{2\text{d}_l \text{Var}(e[2]^n)}\text{Var}(e[2]^n)$$

$$= \frac{1}{2}\text{Var}(x^j) + \frac{1}{2} = \text{Var}(x^j). \tag{3}$$

If we initialize $G_j^i$ to zeros, then its contribution to the variance will increase during training, causing exploding activations in the mainnet. Hence, we prefer to introduce a factor of $1/2$ to divide the variance between the weight and bias generation, where the variance of each component is allowed to either decrease or increase during training. This becomes a problem if the variance of the activations in the mainnet deviates too far away from 1, but we found that it works well in practice.

## 4.2 Hyperfan-out

**Case 1. The hypernet generates the weights but not the biases of the mainnet.** A similar derivation can be done for the backward pass using analogous assumptions on gradients flowing

$$\text{in the mainnet: } \quad \frac{\partial L}{\partial x[t]^{i_t}} = \frac{\partial L}{\partial x[t+1]^{i_{t+1}}} W[t]_{i_t}^{i_{t+1}},$$

$$\text{through mainnet weights: } \quad \frac{\partial L}{\partial W[t]_{i_t}^{i_{t+1}}} = \frac{\partial L}{\partial x[t+1]^{i_{t+1}}} x[t]^{i_t}, \quad \frac{\partial L}{\partial h[t](e)^{k_t}} = \frac{\partial L}{\partial W[t]_{i_t}^{i_{t+1}}} H[t]_{i_t k_t}^{i_{t+1}},$$

$$\text{and through mainnet biases: } \quad \frac{\partial L}{\partial b[t]^{i_{t+1}}} = \frac{\partial L}{\partial x[t+1]^{i_{t+1}}}, \quad \frac{\partial L}{\partial g[t](e)^{l_t}} = \frac{\partial L}{\partial b[t]^{i_{t+1}}} G[t]_{l_t}^{i_{t+1}}. \tag{4}$$

If we initialize the output layer $H$ with the analogous hyperfan-out formula $\text{Var}(H[t]_{i_t k_t}^{i_{t+1}}) = \frac{1}{\text{d}_{i_{t+1}} \text{d}_{k_t} \text{Var}(e^{k_t})}$ and the rest of the hypernet with fan-in init, then we can preserve input and output gradients on the mainnet: $\text{Var}(\frac{\partial L}{\partial x[t]^{i_t}}) = \text{Var}(\frac{\partial L}{\partial x[t+1]^{i_{t+1}}})$. However, note that the gradients will shrink when flowing from the mainnet to the hypernet: $\text{Var}(\frac{\partial L}{\partial h[t](e)^{k_t}}) = \frac{\text{d}_{i_t}}{\text{d}_{k_t} \text{Var}(e^{k_t})}\text{Var}(\frac{\partial L}{\partial W[t]_{i_t}^{i_{t+1}}})$, and scaled by a depth-independent factor due to the use of fan-in rather than fan-out init.

**Case 2. The hypernet generates both the weights and biases of the mainnet.** In the classical case, the forward version (fan-in init) and the backward version (fan-out init) are symmetrical. This

remains true for hypernets if they only generated the weights of the mainnet. However, if they were to also generate the biases, then the symmetry no longer holds, since the biases do not affect the gradient flow in the mainnet but they do so for the hypernet (c.f. Equation 4). Nevertheless, we can initialize $G$ so that it helps hyperfan-out init preserve activation variance on the forward pass as much as possible (keeping the assumption that $\text{Var}(x^j) = 1$ as before):

$$
\begin{aligned}
\text{Var}(y^i) &= \sum_j \big[\text{Var}(W^i_j x^j)\big] + \text{Var}(b^i) \\
&= \mathrm{d}_j \mathrm{d}_k \text{Var}(e[1]^m) \text{Var}(H[\text{hyperfan-out}]^i_{jk}) \text{Var}(x^j) + \mathrm{d}_l \text{Var}(e[2]^n) \text{Var}(G^i_l) \\
&= \mathrm{d}_j \mathrm{d}_k \text{Var}(e[1]^m) \text{Var}(H[\text{hyperfan-in}]^i_{jk}) \text{Var}(x^j)
\end{aligned}
\tag{5}
$$

Plugging in the formulae for Hyperfan-in and Hyperfan-out from above, we get

$$
\implies \text{Var}(G^i_l) = \frac{(1 - \frac{\mathrm{d}_j}{\mathrm{d}_i})}{\mathrm{d}_l \text{Var}(e[2]^n)}.
$$

We summarize the variance formulae for hyperfan-in and hyperfan-out init in Table 1. It is not uncommon to re-use the same hypernet to generate different parts of the mainnet, as was originally done in Ha et al. (2016). We discuss this case in more detail in Appendix Section A.

Table 1: Hyperfan-in and Hyperfan-out Variance Formulae for $W^i_j = H^i_{jk} h(e[1])^k + \beta^i_j$. If $y^i = \text{ReLU}(W^i_j x^j + b^i)$, then $\mathbb{1}_{\text{ReLU}} = 1$, else if $y^i = W^i_j x^j + b^i$, then $\mathbb{1}_{\text{ReLU}} = 0$. If $b^i = G^i_l g(e[2])^l + \gamma^i$, then $\mathbb{1}_{\text{HBias}} = 1$, else if $b^i = 0$, then $\mathbb{1}_{\text{HBias}} = 0$. We initialize $h$ and $g$ with fan-in init, and $\beta^i_j, \gamma^i = 0$. For convolutional layers, we have to further divide $\text{Var}(H^i_{jk})$ by the size of the receptive field. Uniform init: $X \sim \mathcal{U}(-\sqrt{3\text{Var}(X)}, \sqrt{3\text{Var}(X)})$. Normal init: $X \sim \mathcal{N}(0, \text{Var}(X))$.

| Initialization | Variance Formula | Initialization | Variance Formula |
|---|---|---|---|
| Hyperfan-in | $\text{Var}(H^i_{jk}) = \frac{2^{\mathbb{1}_{\text{ReLU}}}}{2^{\mathbb{1}_{\text{HBias}}} \mathrm{d}_j \mathrm{d}_k \text{Var}(e[1]^m)}$ | Hyperfan-out | $\text{Var}(H^i_{jk}) = \frac{2^{\mathbb{1}_{\text{ReLU}}}}{\mathrm{d}_i \mathrm{d}_k \text{Var}(e[1]^m)}$ |
| Hyperfan-in | $\text{Var}(G^i_l) = \frac{2^{\mathbb{1}_{\text{ReLU}}}}{2\mathrm{d}_l \text{Var}(e[2]^n)}$ | Hyperfan-out | $\text{Var}(G^i_l) = \max\left(\frac{2^{\mathbb{1}_{\text{ReLU}}}(1 - \frac{\mathrm{d}_j}{\mathrm{d}_i})}{\mathrm{d}_l \text{Var}(e[2]^n)}, 0\right)$ |

## 5 EXPERIMENTS

We evaluated our proposed methods on four sets of experiments involving different use cases of hypernetworks: feedforward networks, continual learning, convolutional networks, and Bayesian neural networks. In all cases, we optimize with vanilla SGD and sample from the uniform distribution according to the variance formula given by the init method. More experimental details can be found in Appendix Section B.

### 5.1 FEEDFORWARD NETWORKS ON MNIST

As an illustrative first experiment, we train a feedforward network with five hidden layers (500 hidden units), a hyperbolic tangent activation function, and a softmax output layer, on MNIST across four different settings: (1) a classical network with Xavier init, (2) a hypernet with Xavier init that generates the weights of the mainnet, (3) a hypernet with hyperfan-in init that generates the weights of the mainnet, (4) and a hypernet with hyperfan-out init that generates the weights of the mainnet.

The use of hyperfan init methods on a hypernetwork reproduces mainnet weights similar to those that have been trained from Xavier init on a classical network, while the use of Xavier init on a hypernetwork causes exploding activations right at the beginning of training (see Figure 1). Observe in Figure 2 that when the hypernetwork is initialized in the proper scale, the magnitude of generated weights stabilizes quickly. This in turn leads to a more stable training regime, as seen in Figure 3. More visualizations of the activations and gradients of both the mainnet and hypernet can be viewed in Appendix Section B.1. Qualitatively similar observations were made when we replaced the activation function with ReLU and Xavier with Kaiming init, with Kaiming init leading to even bigger activations at initialization.

Suppose now the hypernet generates both the weights and biases of the mainnet instead of just the weights. We found that this architectural change leads the hyperfan init methods to take more time (but still less than Xavier init), to generate stable mainnet weights (c.f. Figure 25 in the Appendix).

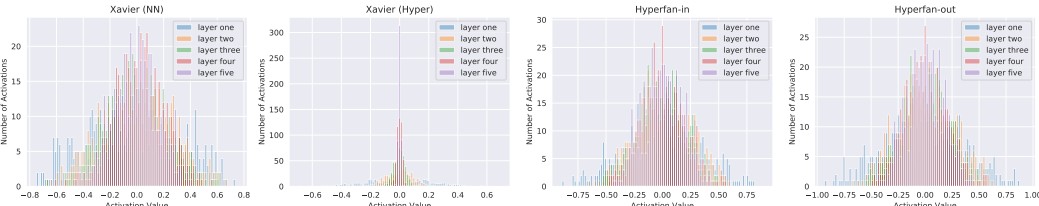

Figure 1: Mainnet Activations before the Start of Training on MNIST.

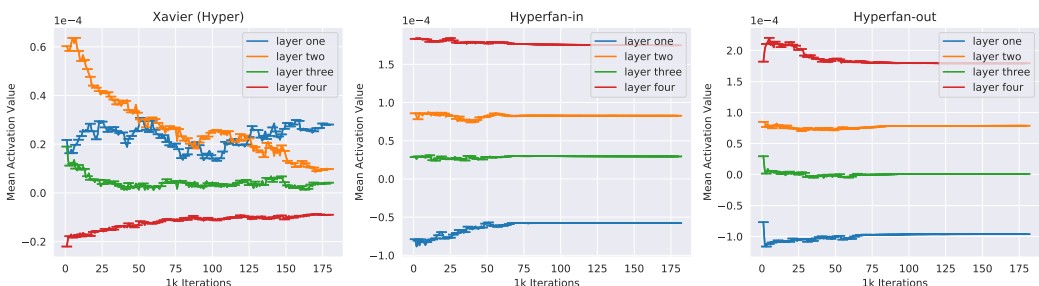

Figure 2: Evolution of Hypernet Output Layer Activations during Training on MNIST. Xavier init results in unstable mainnet weights throughout training, while hyperfan-in and hyperfan-out init result in mainnet weights that stabilize quickly.

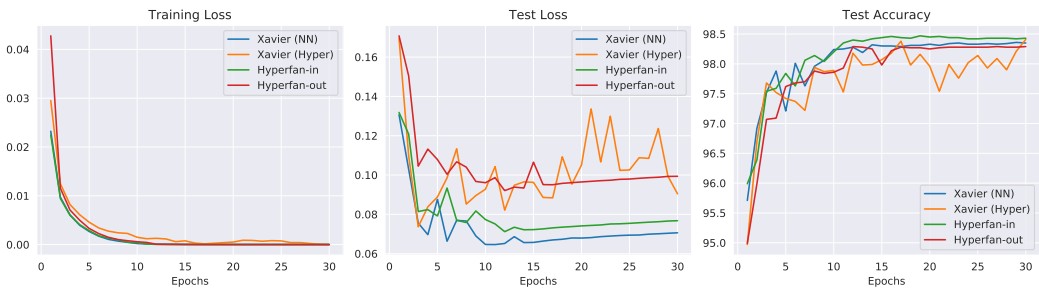

Figure 3: Loss and Test Accuracy Plots on MNIST.

## 5.2 CONTINUAL LEARNING ON REGRESSION TASKS

Continual learning solves the problem of learning tasks in sequence without forgetting prior tasks. von Oswald et al. (2019) used a hypernetwork to learn embeddings for each task as a way to efficiently regularize the training process to prevent catastrophic forgetting. We compare different initialization schemes on their hypernetwork implementation, which generates the weights and biases of a ReLU mainnet with two hidden layers to solve a sequence of three regression tasks.

In Figure 4, we plot the training loss averaged over 15 different runs, with the shaded area showing the standard error. We observe that the hyperfan methods produce smaller training losses at initialization and during training, eventually converging to a smaller loss for each task.

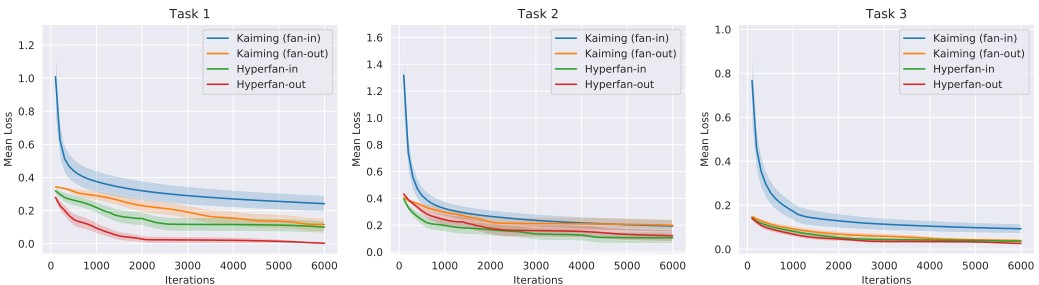

Figure 4: Continual Learning Loss on a Sequence of Regression Tasks.

## 5.3 CONVOLUTIONAL NETWORKS ON CIFAR-10

Ha et al. (2016) applied a hypernetwork on a convolutional network for image classification on CIFAR-10. We note that our initialization methods do not handle residual connections, which were in their chosen mainnet architecture and are important topics for future study. Instead, we implemented their hypernetwork architecture on a mainnet with the All Convolutional Net architecture (Springenberg et al., 2014) that is composed of convolutional layers and ReLU activation functions.

After searching through a dense grid of learning rates, we failed to enable the fan-in version of Kaiming init to train even with very small learning rates. The fan-out version managed to begin delayed training, starting from around epoch 270 (see Figure 5). By contrast, both hyperfan-in and hyperfan-out init led to successful training immediately. This shows a good init can make it possible to successfully train models that would have otherwise been unamenable to training on a bad init.

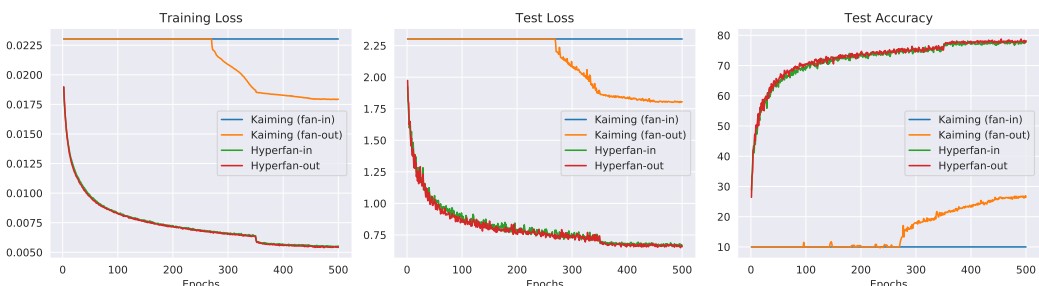

Figure 5: Loss and Test Accuracy Plots on CIFAR-10.

## 5.4 BAYESIAN NEURAL NETWORKS ON IMAGENET

Bayesian neural networks improve model calibration and provide uncertainty estimation, which guard against the pitfalls of overconfident networks. Ukai et al. (2018) developed a Bayesian neural network by using a hypernetwork to simulate an expressive prior distribution. We trained a similar hypernetwork by applying Ukai et al. (2018)'s methods on ImageNet, but differed in our choice of MobileNet (Howard et al., 2017) as a mainnet architecture that does not have residual connections.

In the work of Ukai et al. (2018), it was noticed that even with the use of batch normalization in the mainnet, classical initialization approaches still led to diverging losses (due to exploding activations, c.f. Section 3). We observe similar results in our experiment (see Figure 6) — the fan-in version of Kaiming init, which is the default initialization in popular deep learning libraries like PyTorch and Chainer, resulted in substantially higher initial losses and led to slower training than the hyperfan methods. We found that the observation still stands even when the last layer of the mainnet is not generated by the hypernet. This shows that while batch normalization helps, it is not the solution for a bad init that causes exploding activations. Our approach solves this problem in a principled way, and is preferable to the trial-and-error based heuristics that Ukai et al. (2018) had to resort to in order to train their model.

Surprisingly, the fan-out version of Kaiming init led to similar results as the hyperfan methods, suggesting that batch normalization might be sufficient to correct the bad initializations that result in vanishing activations. That being said, hypernet practitioners should not expect batch normalization to be the panacea for problems caused by bad initialization, especially in memory-constrained scenarios. In a Bayesian neural network application (especially in hypernet architectures without relaxed weight-sharing), the blowup in the number of parameters limits the use of big batch sizes, which is essential to the performance of batch normalization (Wu & He, 2018). For example, in this experiment, our hypernet model requires 32 times as many parameters as a classical MobileNet.

To the best of our knowledge, the interaction between batch normalization and initialization is not well-understood, even in the classical case, and thus, our findings prompt an interesting direction for future research.

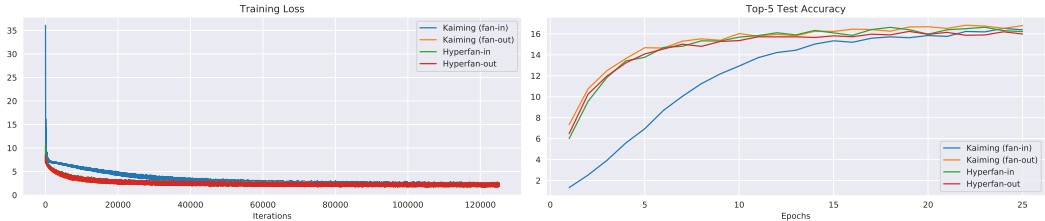

Figure 6: Loss and Test Accuracy Plots on ImageNet.

In all our experiments, hyperfan-in and hyperfan-out both led to successful hypernetwork training with SGD. We did not find a good reason to prefer one over the other (similar to He et al. (2015)'s observation in the classical case for fan-in and fan-out init).

## 6 CONCLUSION

For a long time, the promise of deep nets to learn rich representations of the world was left unfulfilled due to the inability to train these models. The discovery of greedy layer-wise pre-training (Hinton et al., 2006; Bengio et al., 2007) and later, Xavier and Kaiming init, as weight initialization strategies to enable such training was a pivotal achievement that kickstarted the deep learning revolution. This underscores the importance of model initialization as a fundamental step in learning complex representations.

In this work, we developed the first principled weight initialization methods for hypernetworks, a rapidly growing branch of meta-learning. We hope our work will spur momentum towards the development of principled techniques for building and training hypernetworks, and eventually lead to significant progress in learning meta representations. Other non-hypernetwork methods of neural network generation (Stanley et al., 2009; Koutnik et al., 2010) can also be improved by considering whether their generated weights result in exploding activations and how to avoid that if so.

## 7 ACKNOWLEDGEMENTS

This research was supported in part by the US Defense Advanced Research Project Agency (DARPA) *Lifelong Learning Machines* Program, grant HR0011-18-2-0020. We thank Dan Martin and Yawei Li for helpful discussions, and the ICLR reviewers for their constructive feedback.

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

APPENDIX

## A    RE-USING HYPERNET WEIGHTS

### A.1    FOR MAINNET WEIGHTS OF THE SAME SIZE

For model compression or weight-sharing purposes, different parts of the mainnet might be generated by the same hypernet function. This will cause some assumptions of independence in our analysis to be invalid. Consider the example of the same hypernet being used to generate multiple different mainnet weight layers of the same size, i.e. $H[t]_{i_t k}^{i_{t+1}} = H[t+1]_{i_{t+1} k}^{i_{t+2}}, d_{i_{t+1}} = d_{i_{t+2}} = d_{i_t}$. Then, $x[t+1]^{i_{t+1}} = H[t]_{i_t k_t}^{i_{t+1}} e[t]^{k_t} x[t]^{i_t} \not\perp W[t+1]_{i_{t+1}}^{i_{t+2}} = H[t+1]_{i_{t+1} k}^{i_{t+2}} e[t+1]^{k_{t+1}}$.

The relaxation of some of these independence assumptions does not always prove to be a big problem in practice, because the correlations introduced by repeated use of $H$ can be minimized with the use of flat distributions like the uniform distribution. It can even be helpful, since the re-use of the same hypernet for different layers causes the gradient flowing through the hypernet output layer to be the sum of the gradients from the weights of these layers: $\frac{\partial L}{\partial h(e)^k} = \sum_t \frac{\partial L}{\partial W[t]_{i_t}^{i_{t+1}}} H_{i_t k}^{i_{t+1}}$, thus combating the shrinking effect.

### A.2    FOR MAINNET WEIGHTS OF DIFFERENT SIZES

Similar reasoning applies if the same hypernet was used to generate differently sized subsets of weights in the mainnet. However, we encourage avoiding this kind of hypernet architecture design if not otherwise essential, since it will complicate the initialization formulae listed in Table 1.

Consider Ha et al. (2016)'s hypernetwork architecture. Their two-layer hypernet generated weight chunks of size $(K, n, n)$ for a main convolutional network where $K = 16$ was found to be the highest common factor among the size of mainnet layers, and $n^2 = 9$ was the size of the receptive field. We simplify the presentation by writing $i$ for $i_t$, $j$ for $j_t$, $k$ for $k_{t,m}$, and $l$ for $l_{t,m}$.

$$W[t]_j^i = \begin{cases} H_k^{i(\text{mod } K)} \alpha[t][j + \lfloor \frac{i}{K} \rfloor d_j]^k + \beta^{i(\text{mod } K)} & \text{if } i \text{ is divisible by } K \\ \delta_{j(\text{mod } K)j(\text{mod } K)} \left[ H_k^{j(\text{mod } K)} \alpha[t][i + \lfloor \frac{j}{K} \rfloor d_i]^k + \beta^{j(\text{mod } K)} \right] & \text{if } j \text{ is divisible by } K \end{cases}$$

$$\alpha[t][m_t]^k = G[t][m_t]_l^k e[t][m_t]^l + \gamma[t][m_t]^k$$

(6)

Because the output layer $(H, \beta)$ in the hypernet was re-used to generate mainnet weight matrices of different sizes (i.e. in general, $i_t \neq i_{t+1}, j_t \neq j_{t+1}$), $G$ effectively becomes the output layer that we want to be considering for hyperfan-in and hyperfan-out initialization.

Hence, to achieve fan-in in the mainnet $\text{Var}(W[t]_j^i) = \frac{1}{d_j}$, we have to use fan-in init for $H$ (i.e. $\text{Var}(H_k^{i(\text{mod } K)}) = \frac{1}{d_k} \neq \frac{1}{d_j d_k \text{Var}(e[t][m_t]^l)}$), and hyperfan-in init for $G$ (i.e. $\text{Var}(G[t][m_t]_l^k) = \frac{1}{d_j d_l \text{Var}(e[t][m_t]^l)}$).

Analogously, to achieve fan-out in the mainnet $\text{Var}(W[t]_j^i) = \frac{1}{d_i}$, we have to use fan-in init for $H$ (i.e. $\text{Var}(H_k^{i(\text{mod } K)}) = \frac{1}{d_k} \neq \frac{1}{d_i d_k \text{Var}(e[t][m_t]^l)}$), and hyperfan-out init for $G$ (i.e. $\text{Var}(G[t][m_t]_l^k) = \frac{1}{d_i d_l \text{Var}(e[t][m_t]^l)}$).

## B    More Experimental Details

### B.1    Feedforward Networks on MNIST

The networks were trained on MNIST for 30 epochs with batch size 10 using a learning rate of 0.0005 for the hypernets and 0.01 for the classical network. The hypernets had one linear layer with embeddings of size 50 and different hidden layers in the mainnet were all generated by the same hypernet output layer with a different embedding, which was randomly sampled from $\mathcal{U}(-\sqrt{3}, \sqrt{3})$ and fixed. We use the mean cross entropy loss for training, but the summed cross entropy loss for testing.

We show activation and gradient plots for two cases: (i) the hypernet generates only the weights of the mainnet, and (ii) the hypernet generates both the weights and biases of the mainnet. (i) covers Figures 3, 1, 7, 8, 9, 10, 11, 12, 2, 13, 14, 15, and 16. (ii) covers Figures 17, 18, 19, 20, 21, 22, 23, 24, 25, 26, 27, 28, and 29.

The activations and gradients in our plots were calculated by averaging across a fixed held-out set of 300 examples drawn randomly from the test set.

In Figures 1, 8, 9, 11, 12, 13, 14, 16, 18, 20, 21, 23, 24, 26, 27, and 29, the y axis shows the number of activations/gradients, while the x axis shows the value of the activations/gradients. The value of activations/gradients from the hypernet output layer correspond to the value of mainnet weights.

In Figures 2, 7, 10, 15, 19, 22, 25, and 28, the y axis shows the mean value of the activations/gradients, while each increment on the x axis corresponds to a measurement that was taken every 1000 training batches, with the bars denoting one standard deviation away from the mean.

### B.1.1 HYPERNET GENERATES ONLY THE MAINNET WEIGHTS

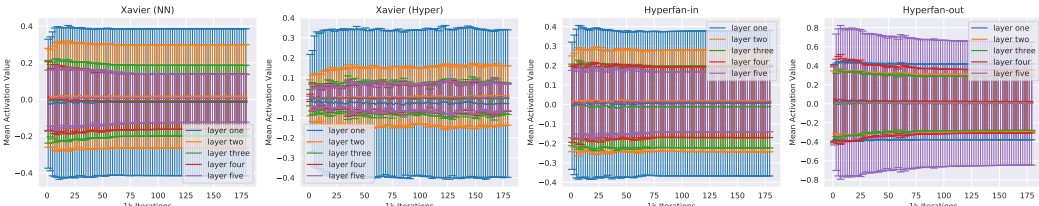

Figure 7: Evolution of Mainnet Activations during Training on MNIST.

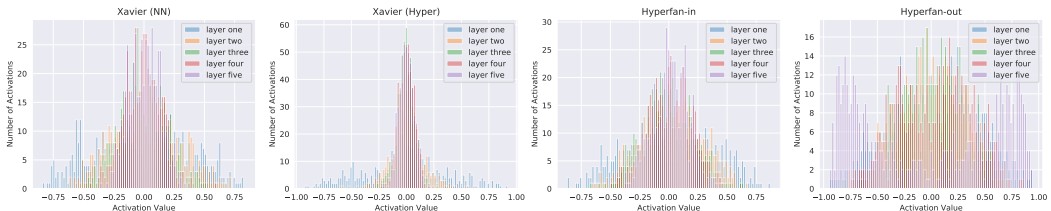

Figure 8: Mainnet Activations at the End of Training on MNIST.

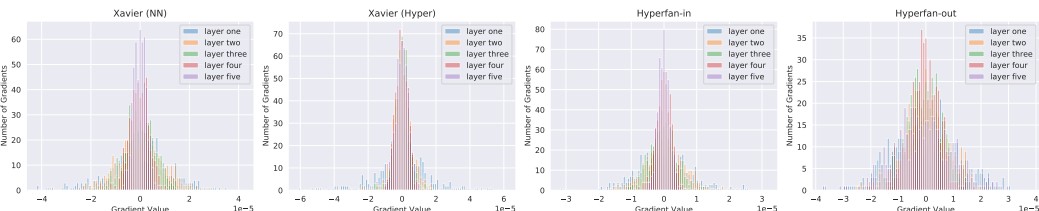

Figure 9: Mainnet Gradients before the Start of Training on MNIST.

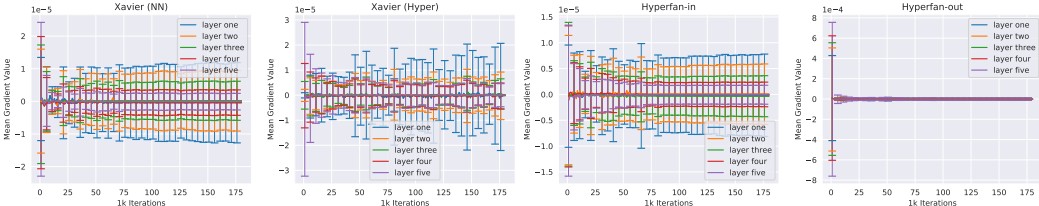

Figure 10: Evolution of Mainnet Gradients during Training on MNIST.

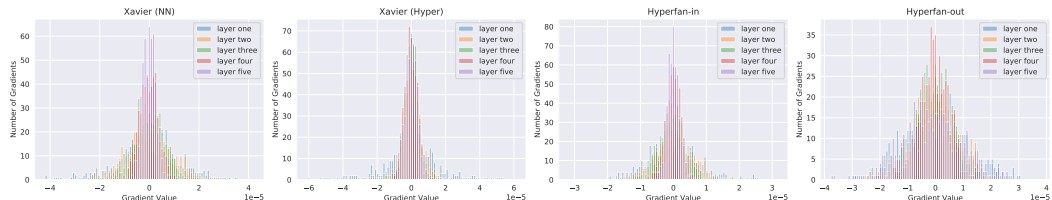

Figure 11: Mainnet Gradients at the End of Training on MNIST.

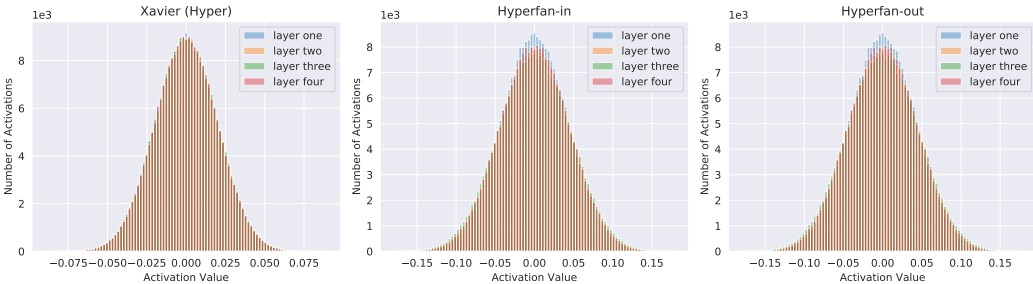

Figure 12: Hypernet Output Layer Activations before the Start of Training on MNIST.

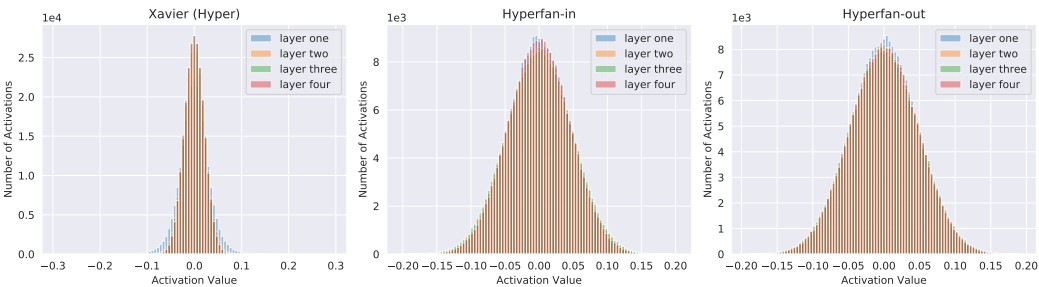

Figure 13: Hypernet Output Layer Activations at the End of Training on MNIST.

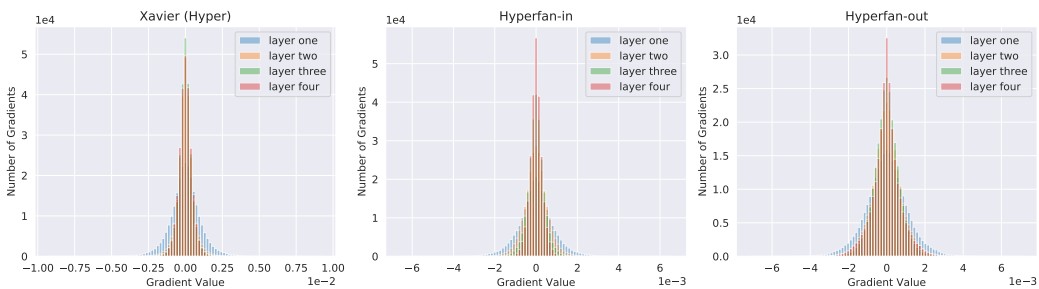

Figure 14: Hypernet Output Layer Gradients before the Start of Training on MNIST.

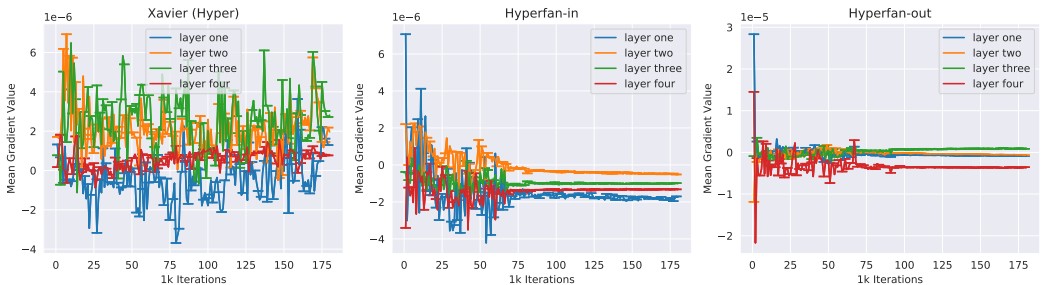

Figure 15: Evolution of Hypernet Output Layer Gradients during Training on MNIST.

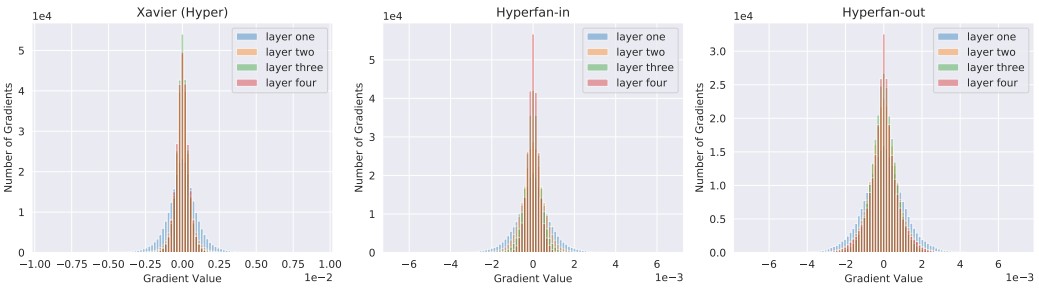

Figure 16: Hypernet Output Layer Gradients at the End of Training on MNIST.

### B.1.2    HYPERNET GENERATES BOTH MAINNET WEIGHTS AND BIASES

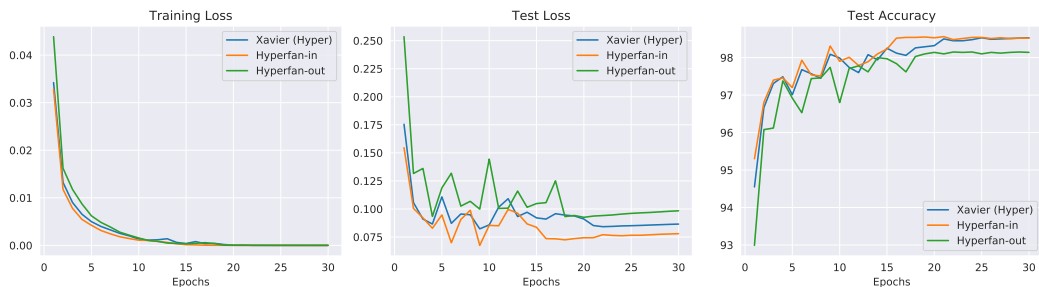

Figure 17: Loss and Test Accuracy Plots on MNIST.

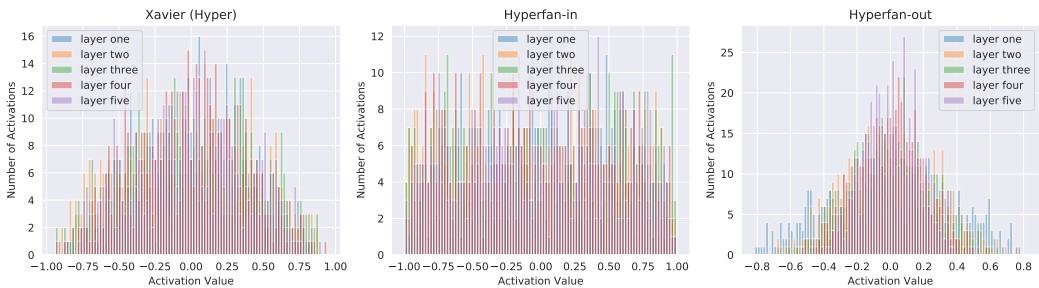

Figure 18: Mainnet Activations before the Start of Training on MNIST.

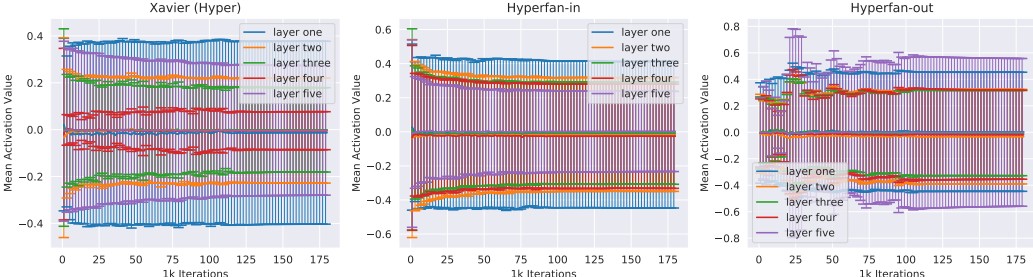

Figure 19: Evolution of Mainnet Activations during Training on MNIST.

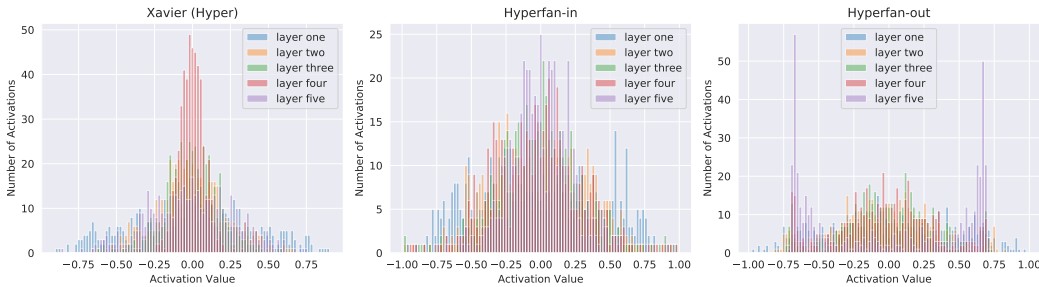

Figure 20: Mainnet Activations at the End of Training on MNIST.

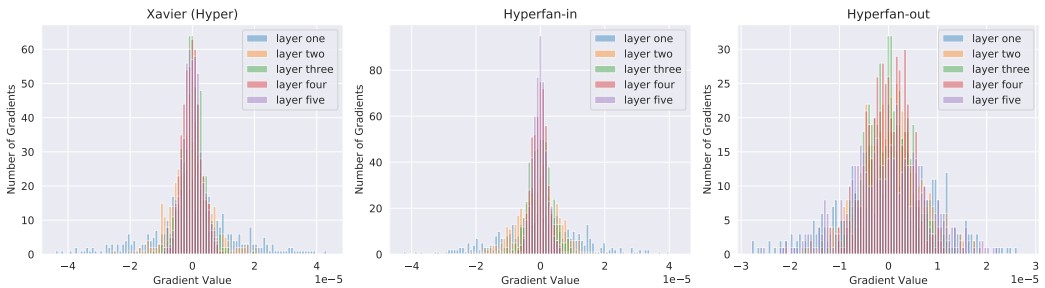

Figure 21: Mainnet Gradients before the Start of Training on MNIST.

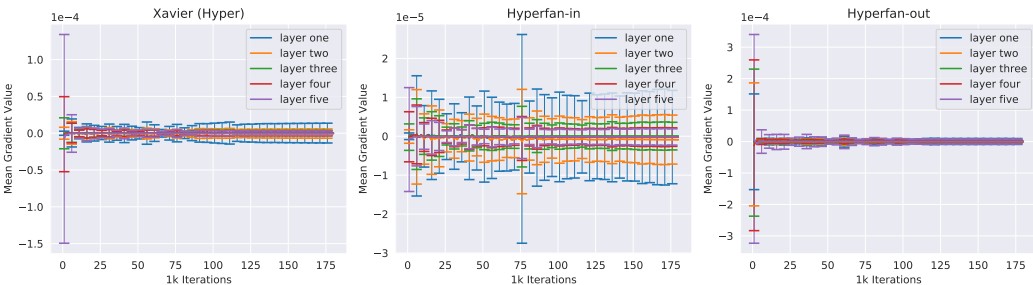

Figure 22: Evolution of Mainnet Gradients during Training on MNIST.

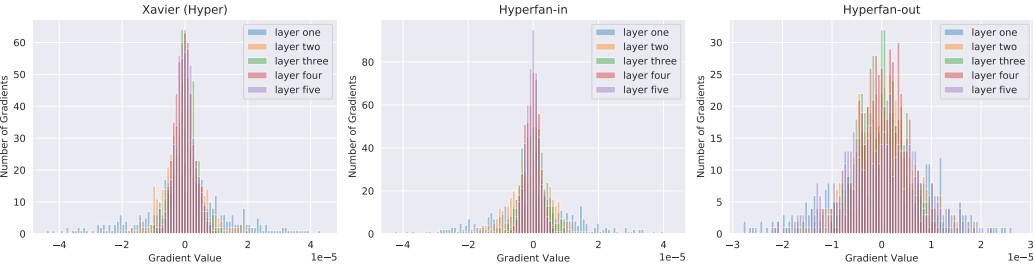

Figure 23: Mainnet Gradients at the End of Training on MNIST.

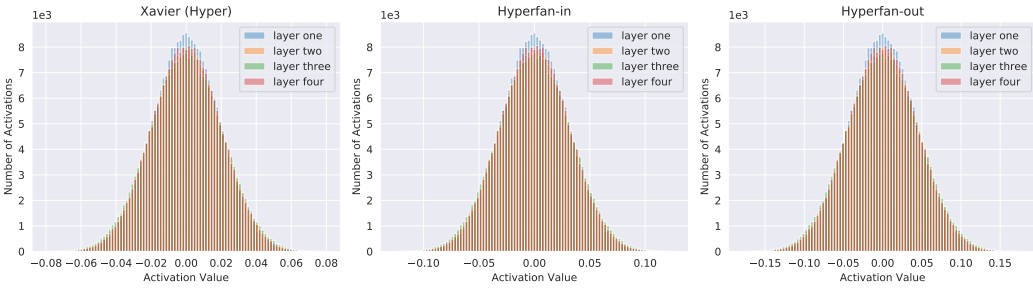

Figure 24: Hypernet Output Layer Activations before the Start of Training on MNIST.

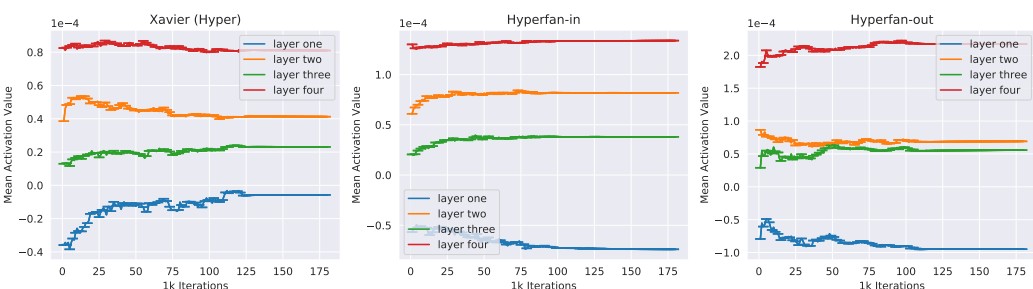

Figure 25: Evolution of Hypernet Output Layer Activations during Training on MNIST.

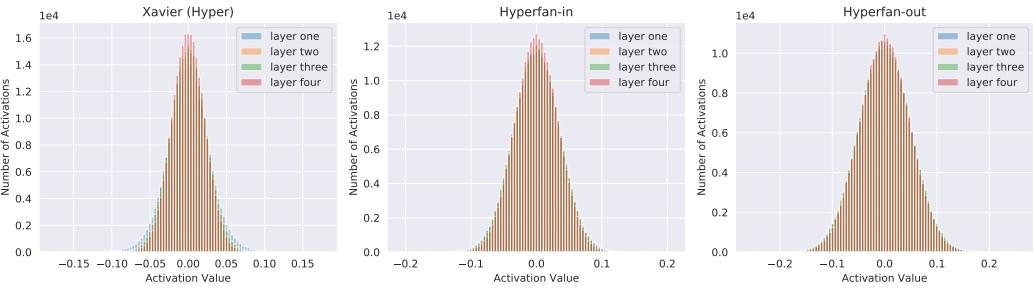

Figure 26: Hypernet Output Layer Activations at the End of Training on MNIST.

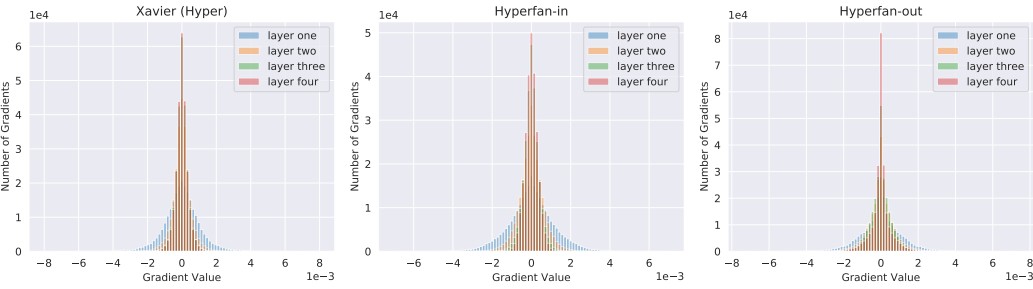

Figure 27: Hypernet Output Layer Gradients before the Start of Training on MNIST.

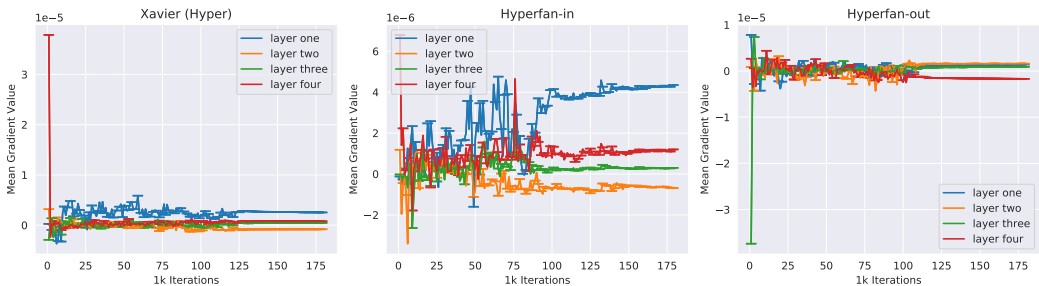

Figure 28: Evolution of Hypernet Output Layer Gradients during Training on MNIST.

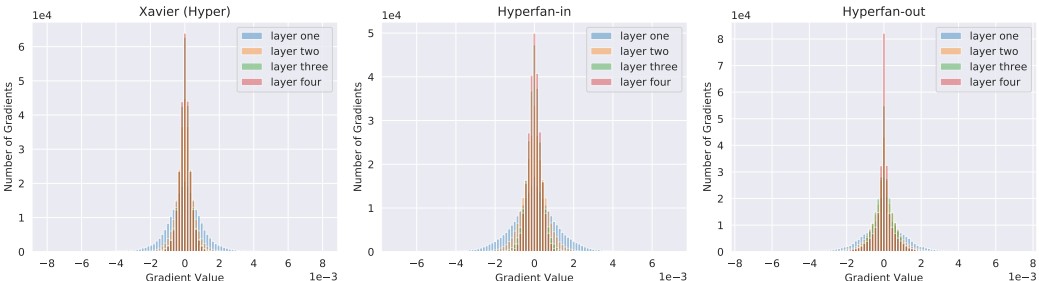

Figure 29: Hypernet Output Layer Gradients at the End of Training on MNIST.

### B.1.3   REMARK ON THE COMBINATION OF FAN-IN AND FAN-OUT INIT

Glorot & Bengio (2010) proposed to use the harmonic mean of the two different initialization formulae derived from the forward and backward pass. He et al. (2015) commented that either version suffices for convergence, and that it does not really matter given that the difference between the two will be a depth-independent factor.

We experimented with the harmonic, geometric, and arithmetic means of the two different formulae in both the classical and the hypernet case. There was no indication of any significant benefit from taking any of the three different means in both cases. Thus, we confirm and concur with He et al. (2015)'s original observation that either the fan-in or the fan-out version suffices.

### B.2   CONTINUAL LEARNING ON REGRESSION TASKS

The mainnet is a feedforward network with two hidden layers (10 hidden units) and the ReLU activation function. The weights and biases of the mainnet are generated from a hypernet with two hidden layers (10 hidden units) and trainable embeddings of size 2 sampled from $\mathcal{U}(-\sqrt{3}, \sqrt{3})$. We keep the same continual learning hyperparameter $\beta_{output}$ value of 0.005 and pick the best learning rate for each initialization method from $\{10^{-2}, 10^{-3}, 10^{-4}, 10^{-5}\}$. Notably, Kaiming (fan-in) could only be trained from learning rate $10^{-5}$, with losses diverging soon after initialization using the other learning rates. Each task was trained for 6000 training iterations using batch size 32, with Figure 4 plotted from losses measured at every 100 iterations.

### B.3   CONVOLUTIONAL NETWORKS ON CIFAR-10

The networks were trained on CIFAR-10 for 500 epochs starting with an initial learning rate of 0.0005 using batch size 100, and decaying with $\gamma = 0.1$ at epochs 350 and 450. The hypernet is composed of two layers (50 hidden units) with separate embeddings and separate input layers but shared output layers. The weight generation happens in blocks of $(96, 3, 3)$ where $K = 96$ is the highest common factor between the different sizes of the convolutional layers in the mainnet and $n = 3$ is the size of the convolutional filters (see Appendix Section A.2 for a more detailed explanation on the hypernet architecture). The embeddings are size 50 and fixed after random sampling from $\mathcal{U}(-\sqrt{3}, \sqrt{3})$. We use the mean cross entropy loss for training, but the summed cross entropy loss for testing.

### B.4   BAYESIAN NEURAL NETWORK ON IMAGENET

Ukai et al. (2018) showed that a Bayesian neural network can be developed by using a hypernetwork to express a prior distribution without substantial changes to the vanilla hypernetwork setting. Their methods simply require putting $\mathcal{L}_2$-regularization on the model parameters and sampling from stochastic embeddings. We trained a linear hypernet to generate the weights of a MobileNet mainnet architecture (excluding the batch normalization layers), using the block-wise sampling strategy described in Ukai et al. (2018), with a factor of 0.0005 for the $\mathcal{L}_2$-regularization. We initialize fixed embeddings of size 32 sampled from $\mathcal{U}(-\sqrt{3}, \sqrt{3})$, and sample additive stochastic noise coming from $\mathcal{U}(-0.1, 0.1)$ at the beginning of every mini-batch training. The training was done on ImageNet with batch size 256 and learning rate 0.1 for 25 epochs, or equivalently, 125125 iterations. The testing was done with 10 Monte Carlo samples. We omit the test loss plots due to the computational expense of doing 10 forward passes after every mini-batch instead of every epoch.

