# OpenReview forum: "Principled Weight Initialization for Hypernetworks"
_ICLR.cc/2020/Conference — Accept (Talk)_

### Official Review · AnonReviewer2 · 2019-10-21
**Official Blind Review #2**

**Rating:** 8

**Review:**

Review of “Principled Weight Initialization for Hypernetworks”

There has been a lot of existing work on neural network initialization, and much of this work has made large impact in making deep learning models easier to train in practice. There has also been a line of work on indirect encoding of neural works (i.e. HyperNEAT work of Stanley, and more recent Hypernetworks proposed by Ha et al) which showed promising results of training very large networks (in the case of Stanley), or have network weights that can adapt to the training data (in the case of Hypernetworks), and these approaches have been shown to be useful in applications such as meta-learning or few-shot learning (i.e. [1]). However, as far as I know, there hasn't been any work that looks at a principled way of initializing the weights of a weight-generating network, which this work tries to explore.

Making the observation (and claim) that traditional init methods don't init hypernetworks properly, they propose a few techniques to initialize hypernetworks ("Hyperfan"-family), which are justified in a similar way as original classical init techniques (i.e. preserving variance like in Xavier init), and they demonstrate that their method works well for feed forward networks on MNIST, CIFAR-10 tasks compared to traditional classical init methods, as well for a continual learning task.

I liked the paper as they identified a problem that hasn't been studied, and proposed a reasonable method to solve it. Their method may be able to make Hypernetworks accessible to many more researchers and practitioners, the way classifical init techniques have made neural net training more accessible.

There are a few things that could improve the paper (and get an improvement score from me). The authors don't have to do all of these, but just a few suggestions:

1) The experiments, to my understanding, are all feed forward networks. How about RNNs or LSTMs?

2) Are there any (interesting) tasks that use Hypernetworks that are not trainable with existing methods, but made trainable using this proposed scheme?

3) Would this method also work with HyperNEAT [2] or Compressed Network Search [3]? (probably should cite that line of work too). In [3], a research group at IDSIA used DCT compression to compress millions of weights into a few dozen parameters, so would be interesting if the approach will work on similar "learn-from-pixels" RL experiments.

I'm assigning a score of 6 (it's currently like a "really good" workshop paper, but a normal conf paper IMO), but I like this paper and would like to see the authors make an attempt to improve it, so I can improve the score to see it get accepted with a higher certainty.

Good luck!

[1] i.e. https://www.ncbi.nlm.nih.gov/pmc/articles/PMC6519722/ https://arxiv.org/pdf/1710.03641.pdf https://arxiv.org/pdf/1703.03400.pdf

[2] http://eplex.cs.ucf.edu/hyperNEATpage/

[3] http://people.idsia.ch/~juergen/compressednetworksearch.html

*** Revised Score ***

Nov20: Upon reading the other reviews, and looking at the changes to the paper with the extra citations, I'm improving the score to 8. (For the record, if this was a 1-10 scale, I would have liked my score to be a 7).

**Experience Assessment:**

I have published one or two papers in this area.

**Review Assessment: Checking Correctness Of Derivations And Theory:**

I did not assess the derivations or theory.

**Review Assessment: Checking Correctness Of Experiments:**

I assessed the sensibility of the experiments.

**Review Assessment: Thoroughness In Paper Reading:**

I read the paper at least twice and used my best judgement in assessing the paper.

---

> ### Author Response · Authors · 2019-11-14
> **New Bayesian Neural Network task introduced in Section 5.4**
>
> We would like to thank the reviewer for his thoughtful feedback on our work.
>
> Regarding the reviewer’s questions:
>
> 1) Variance analysis does not extend to RNNs and LSTMs even in the classical setting without the use of hypernetworks. We believe this is an important open problem that deserves attention and we leave it for future work.
>
> 2) Hypernetworks are a natural choice for enabling Bayesian neural networks, because they can be used to simulate an expressive prior distribution. This is useful for improving model calibration, providing uncertainty estimation, and doing approximate Bayesian inference or regularization. [1] demonstrated a method to build a Bayesian network using a hypernet, but they mentioned having to use a heuristic method to initialize the weights of their hypernetwork in order to prevent losses from diverging.
>
> [1]’s implementation was unfortunately not publicly available so we could not try our approach on their exact settings. We introduced a similar experiment in Section 5.4 that differed with [1]’s in terms of residual connections in the mainnet.  While all initializations led to trainable networks in this case, our proposed initialization gave superior performance compared to the default initialization in PyTorch. We are confident that an extension of our approach to hypernetworks with residual mainnets should be able to resolve [1]’s initializations issues. This work is an important stepping stone towards this direction.
>
> 3) Non-hypernetwork methods like HyperNEAT or Compressed Network Search that generate the weights of some target network would probably also do well to consider if their generated weights might result in exploding activations. Because most of this line of work was done before the advent of deep learning, the networks considered were most likely not deep enough to have caused an issue. We hope that our paper will drive new progress in this line of research as well. We have cited the papers mentioned, thanks for pointing them out.
>
> [1] Hypernetwork-based Implicit Posterior Estimation and Model Averaging of Convolutional Neural Networks. Ukai et al. ACML 2018.

---

### Official Review · AnonReviewer3 · 2019-10-22
**Official Blind Review #3**

**Rating:** 8

**Review:**

The paper presents an extension of Glorot/He weight variance initiazation formula for the hypernetworks. Hypernetworks are the class of neural networks where one (hyper) model generates the weights for the another (main) network, introduced by Ha et.al in 2016.
Authors argue and show via experiments that standard weight init formulas do not work for hypernetworks, resulting in vanishing or exploding activations and re-derive the formula for convolutional/fully-connected networks + ReLU.
They show that proposed method allows  hypernet training when the standard ways don`t.

The technical contribution seems as logical and straightforward yet necessary step for hypernetwork-related research.

Questions:

 - In standard NNs, initialization issues are mostly solved after introduction of BatchNorm. Wouldn`t it be the case for hypernetwork as well: to just add BN layers between main net layers?

 - Figure 2. What are is the y axis of the figure? Norm? Variance? Mean? The same question for the most of Figures in appendix

- It would be nice to see how proposed method stands vs mentioned heuristics like M3 and M4.

Overall I like the paper.

Minor comments:


 > "This fundamental difference suggests that conventional knowledge about neural networks may not
apply directly to hypernetworks and radically new ways of thinking about weight initialization,
optimization dynamics and architecture design for hypernetworks are sorely needed."

I don`t see anything "radically new" in re-derivation of Xavier formula to the new type of network.


======
Revision:

Revised version addressed my concerns and the batchnorm-related experiment is indeed surprising.
Overall, I like the paper and increase evaluation to strong accept.



**Experience Assessment:**

I have read many papers in this area.

**Review Assessment: Checking Correctness Of Derivations And Theory:**

I assessed the sensibility of the derivations and theory.

**Review Assessment: Checking Correctness Of Experiments:**

I carefully checked the experiments.

**Review Assessment: Thoroughness In Paper Reading:**

I read the paper thoroughly.

---

> ### Author Response · Authors · 2019-11-14
> **Batchnorm is very useful, but does not always correct a bad init**
>
> Thank you for your insightful comments.
>
> In response to your questions:
>
> - Batch normalization is an indispensable part of modern neural architectures and goes a long way towards correcting inappropriately scaled activations and gradients. That said, because the normalization happens over the batch dimension, it does not always manage to salvage a bad initialization. For example, [1] remarked that they had to resort to a heuristic method of scaling the hypernetwork weights in order to prevent their loss from diverging, even though their mainnet is already batch normalized.  We added a new experiment in Section 5.4 which uses a batch normalized mainnet, and showed that batch normalization might help correct bad inits with vanishing activations, but not exploding activations. This is a surprising finding. To the best of our knowledge, the interactions between batch normalization and initialization are not well-understood, even in the classical case, and our findings prompt an interesting direction for future research.
>
> - The fan-in version of Kaiming init, which we show causes exploding activations in Section 3, is the default initialization in many deep learning libraries like PyTorch and Chainer. We showed that our proposed initialization scheme resulted in substantially lower losses and faster training, despite the presence of batch normalization.
>
> - In Figure 2, the y axis shows the mean activation value of the hypernet’s output layer (i.e. the mainnet weights), with the bars denoting one standard deviation away from the mean. We added axis labels to all the figures in the paper, and added additional descriptions in the appendix. Thanks for pointing this out.
>
> - The mentioned heuristics were tuned by trial and error for specific architectures and will not work in general. We mention them as examples of ad-hoc ways people initialize hypernets in practice. The principled initialization formulae proposed in this work provide sensible defaults, and thus obviates the need to resort to trial-and-error, saving developers both time and compute.
>
> - We removed the word “radically”.
>
> [1] Hypernetwork-based Implicit Posterior Estimation and Model Averaging of Convolutional Neural Networks. Ukai et al. ACML 2018.

---

> > ### Comment · AnonReviewer3 · 2019-11-15
> > **Re:**
> >
> > Thank you!
> >
> > Revised version addressed my concerns and the batchnorm-related experiment is indeed surprising.
> > Could you please also add test-accuracy graph for the ImageNet experiment?
> >
> > Overall, I like the paper and increase evaluation to strong accept.

---

> > > ### Author Response · Authors · 2019-11-15
> > > **Thanks!**
> > >
> > > Thank you for raising the score.
> > >
> > > Bayesian neural networks consume a lot more compute in the testing phase, and we left them out given the time constraints of the rebuttal period. But we will update the paper with the test graphs in the final manuscript.

---

### Official Review · AnonReviewer1 · 2019-10-22
**Official Blind Review #1**

**Rating:** 8

**Review:**

Principled Weight Initialization for Hypernetworks

Summary:

This paper investigates initialization schemes for hypernets, models where the weights are generated by another neural network, which takes as input either a learned embedding or the activations of the model itself. Standard initialization schemes for deep networks (Glorot, He, Fixup) are based on variance analyses that aim to keep the scale of activations/gradients well-behaved through forward-backward propagation, but using this approach is ill-founded for hypernets where the output is a set of weights rather than e.g. a softmax’d classification output. This paper extends the standard variance analysis to consider the hypernet case by investigating what the choice of hypernet initialization should be if one still wishes to maintain the well-behaved activations/gradients in the main model. The authors present results showing the evolution of hypernet outputs with their scheme are better-behaved, demonstrate that their modification leads to improved stability for hypernet-based convnets on CIFAR  (over a model which, for the standard choice of He init, basically does not train) and improved performance on a continual learning task.

My take:

This is an "aha!" or an “obvious in retrospect” paper: a simple idea based on noticing something being done wrong in practice with a fairly straightforward fix, coupled with a decent empirical evaluation and analysis. The paper is well-written and reasonably easy to follow (although I am not familiar with Ricci calculus I did not feel that I was flummoxed at any point during the maths), and the potential impact is decent: any future work which employs a hypernetwork would likely do well to consider the methods in this paper. While I would like to see the empirical evaluations taken a bit further, I think this paper is a solid accept (7/10) and would make a good addition to ICLR2020.

Notes:

While this is a good paper, I think the impact of the paper could be magnified if the authors were a bit more ambitious with their empirical evaluations. This method seems to enable training hypernets in settings which would previously have been unstable; it would be good to more thoroughly characterize robustness to hyperparameters or otherwise demonstrate additional practical value. Showing results on ImageNet would also be helpful (I’m well aware this is not always in the realm of compute possibility) or just showing progress on a more challenging task outside of CIFAR or a small continual learning problem would, I think, greatly increase the chances that this paper catches on. As this is a “unleash your potential” note, I have not taken this sentiment into account in my review (as should hopefully be evident from my accept score).

Minor:

-Caption for figure 2 should indicate what kind of magnitudes are represented—are these the average weight norms in each layer vs epochs? The axes should be labeled.



**Experience Assessment:**

I have published one or two papers in this area.

**Review Assessment: Checking Correctness Of Derivations And Theory:**

I carefully checked the derivations and theory.

**Review Assessment: Checking Correctness Of Experiments:**

I carefully checked the experiments.

**Review Assessment: Thoroughness In Paper Reading:**

I read the paper thoroughly.

---

> ### Author Response · Authors · 2019-11-14
> **New ImageNet Experiment, Found interesting interaction between initialization and batchnorm**
>
> Many thanks for your encouraging feedback and helpful comments.
>
> We show new results on a Bayesian neural network experiment on ImageNet in Section 5.4. This new experiment shows interesting findings about the interaction between batch normalization and initialization, which is not well-understood even in the classical case, to the best of our knowledge. Contrary to popular belief, batch normalization is not always sufficient to correct a bad init. In our experiment, we observed that the fan-in version of Kaiming init, the default initialization in many deep learning libraries like PyTorch and Chainer, performs significantly worse than the proposed initialization.
>
> We also found that in the case of vanishing activations, batch normalization was able to mitigate the problem. It is not clear why this happens, even in the classical case that does not involve hypernetworks, but our experimental findings prompt an interesting direction for future research. This direction of research is even more important for hypernetworks as the large batch sizes required by batch normalization might be prohibitive for applications in Bayesian neural networks (see Section 5.4 for details).
>
> In Figure 2, the y axis shows the mean activation value of the hypernet’s output layer (i.e. the mainnet weights), with the bars denoting one standard deviation away from the mean. We added axis labels to all the figures in the paper, and added additional descriptions in the appendix. Thanks for pointing this out.

---

### Author Response · Authors · 2019-11-14
**Thank you for the useful suggestions and feedback!**

Thank you to all the reviewers for their feedback. Our main changes include a new experiment in Section 5.4 involving a Bayesian neural network on ImageNet, as well as axis labels for all the figures in the paper. We have found the comments heartening and thoughtful, and will further improve our manuscript based on the feedback.

---

### Author Response · Authors · 2020-09-14
**Example Code**

I have put up some example code in a gist here:
https://gist.github.com/crazyoscarchang/c9a11b67c420202da1f26e0d20786750

Hope this helps for readers who might be confused by the notation in the paper.

---

### Decision · Program_Chairs · 2019-12-19

**Decision:**

Accept (Talk)

**Comment:**

All the reviewers agreed that this was a sensible application of mostly existing ideas from standard neural net initialization to the setting of hypernetworks.  The main criticism was that this method was used to improve existing applications of hypernets, instead of extending their limits of applicability.